# Role of Chondroitin Sulfate Proteoglycan 5 in Steroid-Induced Cataract

**DOI:** 10.3390/cells12131705

**Published:** 2023-06-23

**Authors:** Woong-Sun Yoo, Hyemin Seong, Chieun Song, Mee-Young Choi, Bina Lee, Youngsub Eom, Hae-Jin Kim, Seung Pil Yun, Seong-Jae Kim

**Affiliations:** 1Department of Ophthalmology, Gyeongsang National University College of Medicine, and Gyeongsang National University Hospital, Jinju 52727, Republic of Korea; 2Institute of Health Sciences, Gyeongsang National University College of Medicine, Jinju 52727, Republic of Korea; 3Department of Pharmacology and Convergence Medical Science, Gyeongsang National University College of Medicine, Jinju 52727, Republic of Korea; 4Department of Ophthalmology, Korea University College of Medicine, Seoul 02842, Republic of Korea; 5Department of Ophthalmology, Korea University Ansan Hospital, Ansan 15355, Republic of Korea; 6School of Mechanical and Aerospace Engineering, Gyeongsang National University, Jinju 52828, Republic of Korea

**Keywords:** epithelial–mesenchymal transition, lens epithelial cells, steroid-induced, glucocorticoid treatment, EGF-like domain

## Abstract

Steroid-induced cataracts (SIC) are defined as cataracts associated with the administration of corticosteroids. Long-term glucocorticoid treatment for inflammatory diseases reportedly increases the risk of SIC, and steroids can induce cataracts by disrupting ocular growth factor balance or homeostasis. In this study, we verified the effect of chondroitin sulfate proteoglycan 5 (CSPG5) using dexamethasone (dexa)-treated human lens epithelial (HLE-B3) cells and the lens epithelium from the anterior capsule of SIC patients obtained during cataract surgery. CSPG5 expression increased in the lens epithelium of SIC patients. The downregulation of CSPG5 suppressed the dexa-induced epithelial–mesenchymal transition (EMT)-related protein expression and motility in HLE-B3 cells. The disruption of the transcription factors *EZH2* and *B-Myb* downregulated CSPG5, dexa-induced fibronectin expression, and cell migration in HLE-B3 cells, reaffirming that CSPG5 expression regulates EMT in lens epithelial cells. Taken together, these results indicate that the steroid-induced effects on lens epithelial cells are mediated via alterations in CSPG5 expression. Therefore, our study emphasizes the potential of CSPG5 as a therapeutic target for the prevention and treatment of SIC.

## 1. Introduction

Cataracts cause vision loss in more than 30% of people worldwide [1]. Cataract surgery can improve the vision of patients but incurs financial costs and several side effects, such as dry eye or cystoid macular edema [2,3]. Well-known risk factors for cataracts are age, diabetes, trauma, exposure to ultraviolet radiation, use of drugs such as steroids, and inflammatory eye diseases such as uveitis [4]. Among these, steroid-induced cataract (SIC) is a side effect of the long-term administration of corticosteroids [5]. With the improvement in average life expectancy and rise in the prevalence of diseases that warrant corticosteroid treatment, the incidence of SIC is expected to increase [5,6]. Reportedly, the risk of SIC has increased because of current treatment approaches for inflammatory disorders, such as allergy and rheumatic diseases. Unlike diabetic and age-related cataracts (ARC), SIC can occur at an early age, and vision may deteriorate at an early stage because of the location of lesions [5,6]. However, studies on the exact mechanism and major effects of steroids on the lens are insufficient. Therefore, understanding the changes in the lens caused by steroids is necessary for the prevention and treatment of SIC.

SIC is associated only with steroids which have glucocorticoid activity, such as hydrocortisone and prednisolone, with a dose- and time-dependent manner [5]. In addition, cataract development can be affected by glucocorticoid interaction with the non-specific hepatic glucocorticoid receptors present in the lens and with growth factors involved in both the proliferation of lens epithelial cells (LEC) and regulation of differentiation into fiber cells [5,7,8]. Proteoglycans regulate diverse cellular processes, and chondroitin sulfate proteoglycan reportedly induces the formation of lens vesicles in the chick embryo, associated with epithelial invagination and fusion at an early stage of lens morphogenesis [9,10,11]. Chondroitin sulfate proteoglycan-5 (CSPG5) is a transmembrane protein that is abundantly expressed on the surface of nerve and glial cells, primarily required for the formation of the central nervous system; however, *CSPG5* mRNA and protein expression were discovered in the retina and retinal pigment epithelium in the Leber congenital amaurosis model [12,13,14,15,16]. Our previous work revealed that nerve growth factor (NGF) remarkably suppressed the increased activation of cell migration in LEC that were exposed to high doses of dexamethasone (dexa) [8]. We also reported that NGF inhibits the activation of protein kinase B and p38 mitogen-activated protein kinase, which are activated by dexa in LEC [7]. Notably, *CSPG5* mRNA was also detected in the lens, apart from the brain and retina [10]. However, CSPG5 expression and function in the human lens epithelium and its role in the development of SIC have not been elucidated.

In this study, we aimed to identify the presence of CSPG5 in human lens samples, as well as the relationship between CSPG5 expression and cataract development.

## 2. Materials and Methods

### 2.1. Acquisition of Anterior Lens Capsules

Patients of cataracts were recruited from the Department of Ophthalmology, Gyeongsang National University Hospital for this study. All patients gave informed consent. This study was approved by the institutional review board of Gyeongsang National University Hospital (GNUH2019-05-014). All procedures involving human material were performed in accordance with the current ethical standards of the institutional and national research committees, the Declaration of Helsinki, and its later amendments or comparable ethical standards. Anterior lens capsules of cataract patients were collected during phacoemulsification by a single surgeon (SJK). Continuous circular capsulorhexis was conducted using forceps of 5.0–5.5 mm in diameter. The capsules were rinsed with a balanced salt solution to remove viscoelasticity and collected under sterile conditions.

### 2.2. Definition of Cataract

Trained ophthalmologists used the lens opacity classification system (LOCS) III to classify opacities into six nuclear colors (NC1–NC6), six opalescence (NO1–NO6), five cortical (C1–C5), and five posterior subcapsular (P1–P5) grades of increasing severity, according to photographic standards. ARC is defined as a cataract occurring over 40 years of age, over NO2 and NC2, and under P3 according to LOCS III, without diabetes mellitus, glaucoma, shallow anterior chambers, uveitis, high myopia (axial length <27.0 mm), pseudoexfoliation, traumatic cataract, subluxated cataract, previous ocular surgery, ocular disease (i.e., retinitis pigmentosa), or systemic/intraocular steroid use. SIC is defined as a cataract associated only with systemic or topical steroids that possess glucocorticoid activity and the migration of aberrant LEC from the lens equator to the posterior lens pole, located in a sharply defined, limited central area of the posterior subcapsular region, as previously reported [5].

### 2.3. Human LEC Culture

The anterior capsule of the human lens was extracted into an e-tube or a cell culture medium in the operating room and rinsed in PBS for further analysis. For the primary human LEC culture, a monolayer of the lens epithelium was peeled off, cut into small pieces, and placed at the bottom of 6-well plates for attachment. The medium was changed once per week. This procedure was also approved by the institutional review board of Gyeongsang National University Hospital (GNUH2019-05-014).

### 2.4. Cell Culture and Treatment

HLE-B3 cells, human corneal epithelial cells, and human adult retinal pigment epithelial (ARPE)-19 cells were purchased from the American Type Culture Collection (Rockville, MD, USA). Cells were sustained and grown to ~80–90% confluency, in Minimum Essential Medium (MEM; Cat # 11095080; Thermo Fisher Scientific, Waltham, MA, USA) containing 1% antimicrobial solution and 10% fetal bovine serum (Sigma-Aldrich, St. Louis, MO, USA) in 5% CO_2_ at 37 °C. Cells were treated with 0.1 mg/mL dexa at the specified time points and further treated with 10 μM EI1 and A490, where indicated.

### 2.5. Measurement of Cell Viability

HLE-B3 cells were cultured into 24-well plates at a density of 5 × 10^4^ cells/well. After 24 h, the cells were incubated in a medium with dexa (0.1 mg/mL) for 48 h. Cell viability was measured using the CCK-8 kit (Dojindo Laboratories, Kumamoto, Japan). Cells were incubated in 10 μL of CCK-8 solution for 1 h at 37 °C in a humidified atmosphere containing 5% CO_2_. The amount of formazan dye generated by cellular dehydrogenases was confirmed by measuring absorbance at 450 nm using a microplate reader (Molecular Devices, Sunnyvale, CA, USA). Cells were observed using inverted microscope (Olympus, Tokyo, Japan) to detect phenotypic differences between the treated and control cells.

### 2.6. shRNA Transfection

HLE-B3 cells were seeded into 6-well plates at a density of 4 × 10^6^ cells/well and incubated for 24 h. Cells were transfected with *CSPG5* shRNA or *control* shRNA (SHCLNG-NM013884; Sigma-Aldrich), according to the manufacturer’s instructions. The empty vector was transfected into cells using Lipofectamine 3000 and Opt. MEM^TM^ medium (Life Technologies, Carlsbad, CA, USA). Cells were further processed 24 h after transfection.

### 2.7. Wound Healing Assay

HLE-B3 cells were seeded into 6-well plates at a density of 1 × 10^6^ cells/well. After 24 h, the monolayer was scratched with a sterile 200 μL pipette tip. Cells were then treated with 0.1 mg/mL dexa. In some experiments, they were co-treated with 10 μM EI1 and A490. Cell migration was observed after 8 h, using an inverted microscope (Olympus).

### 2.8. RT^2^ profiler PCR array

First-strand cDNA was obtained from 0.5 μg of each RNA sample, using the RT^2^ First Strand Kit (Qiagen, Hilden, Germany), according to the manufacturer’s instructions. This was mixed with an appropriate RT^2^ SYBR^®^ Green Master Mix and aliquoted into the wells of an RT^2^ Profiler PCR array. Following PCR, relative expression was determined via the ΔΔCT method.

### 2.9. Cytoplasmic and Nuclear Protein Extraction

Nuclear and cytoplasmic extracts were prepared using NE-PER Nuclear and Cytoplasmic Extraction Reagents (Thermo Fisher Scientific, Waltham, MA, USA), according to the manufacturer’s protocol. The medium was removed, and cells were washed with ice-cold PBS. Cell pellets were re-suspended in cold CER I buffer. After a 20 min incubation on ice, the cytoplasmic extract was separated by centrifugation in 13,600 × *g* for 10 min at 4 °C. The pellet was washed with ice-cold PBS and re-suspended in CER II, protease inhibitors, and phosphatase inhibitors. Following a 30 min incubation on ice, the nuclear extract was separated by centrifugation at 13,600 × *g* for 20 min at 4 °C. β-Actin (1:10,000, Sigma-Aldrich) was used as a loading control, α-tubulin (1:10,000, Sigma-Aldrich) as the cytoplasmic protein marker, and lamin A/C (1:1000, Santa Cruz, CA, USA) as the nuclear protein marker.

### 2.10. Immunofluorescence

Extracted anterior capsules of the human lens were placed flat for 7 days in poly-D-lysine coated 24-well plates. They were fixed in 4% paraformaldehyde (Sigma-Aldrich) and blocked using 5% normal donkey serum for 2 h, prior to incubation with anti-CSPG5 (1:500, Abcam, Cambridge, UK) at 4 °C. Subsequently, the tissues were incubated with donkey anti-rabbit immunoglobulin G secondary antibodies and imaged using fluorescence microscopy (Olympus). Data were analyzed using Olympus FLUOVIEW Fv10 Asw 4.2 software, Olympus, Tokyo, Japan.

### 2.11. Quantitative Real-Time PCR

Total RNA was prepared using TRIzol (Invitrogen Life Technologies, Carlsbad, CA, USA), following the manufacturer’s instructions, and 1 μg was used for cDNA synthesis with an iCycler thermocycler (Bio-Rad Laboratories, Hercules, CA, USA). qRT-PCR was performed using an iQ SYBR Green Supermix kit (Bio-Rad Laboratories) with a Light Cycler 480 II (Roche Life Science, Indianapolis, IN, USA). PCR primers were constructed based on the reported cDNA sequences from the NCBI data bank and are listed in Appendix A.

### 2.12. Western Blot Analysis

Cells were lysed in RIPA lysis buffer supplemented with Halt phosphatase inhibitor cocktail (Thermo Fisher Scientific) and Halt protease inhibitor cocktail (Thermo Fisher Scientific). Cell lysates were sonicated and centrifuged at 12,000 × *g* for 10 min at 4 °C to remove insoluble debris. Protein concentrations in the cell lysates were measured using a BCA protein assay kit (Pierce, Rockford, IL, USA). Whole-cell lysates (15–20 µg) were separated by SDS-PAGE on a 10% polyacrylamide gel and transferred to a nitrocellulose membrane (Millipore, Bedford, MA, USA). After blocking with 5% non-fat dry milk, the blots were reacted with primary antibodies against CSPG5 (1:1000, Santa Cruz), fibronectin, vimentin, α-SMA, EZH2, and B-Myb (1:1000, Abcam) and β-actin (1:10,000, Sigma-Aldrich). Next, the membranes were incubated with horseradish peroxidase-conjugated anti-rabbit immunoglobulin (Ig) G or anti-mouse IgG (Cell Signaling Technology, Inc., Danvers, MA, USA). Antibody bonded blot was detected using the SuperSignal Chemiluminescent Substrate (Pierce). Images were acquired using a ChemiDoc Touch Imaging System (Bio-Rad, Hercules, CA, USA). Detection of density was measured using ImageJ (NIH, Bethesda, MD, USA).

### 2.13. Chondroitinase ABC

Proteoglycan cell lysate (50 μg) was incubated with three volumes of 95% ethanol and 1.3% potassium acetate on ice. The lysates were separated by centrifugation at 10,000× *g* for 10 min at 4 °C. Precipitates were dissolved in a chondroitinase reaction buffer (Sigma-Aldrich) and digested with protease-free chondroitinase ABC (CHase ABC; 1 U/50 μL, in PBS with 1% BSA, Sigma-Aldrich). First, proteins were treated with CHase ABC and were precipitated again from the reaction buffer by adding three volumes of the ethanol and potassium acetate solution. Pellets were dissolved in 2× sample buffer and then boiled for 3 min at 100 °C. After cooling to 4 °C, samples were analyzed with western blotting (SDS-PAGE).

### 2.14. Statistical Analysis

All data were analyzed using GraphPad Prism 6. Quantifiable results are suggested as the mean ± S.E.M or standard deviation of over three independent experiments. Statistical significance was confirmed using an unpaired two-tailed Student’s *t*-test or one-way ANOVA test followed by Tukey’s multiple comparisons test. Statistical significance was determined at *p* < 0.05.

## 3. Results

### 3.1. Clinical Characteristics of Cataract Patients

Anterior capsules were obtained from 50 eyes of ARC and 12 eyes of SIC patients. There was no difference in sex, visual acuity, and intraocular pressure for sample acquisition; SIC patients were significantly younger than ARC patients. LOCS III showed significantly higher cortical grades in the ARC and higher posterior subcapsular grades in the SIC (Table 1, Appendix A).

### 3.2. Dexa-Induced Upregulation of CSPG5 Expression

To reveal the effect of dexa, changes in the expression of growth factor genes were analyzed by the RT2 profiler PCR array (Figure 1A). Several genes, such as *SLCO1A2, FIGF, NODAL, IL12B, FGF11, INHA*, and *CSPG5*, were upregulated by dexa treatment in HLE-B3 cells. In particular, *CSPG5* expression showed a significant increase over 72 h (Figure 1B). To analyze the inherent expression of CSPG5, without treatment, protein levels in the HLE-B3 and ARPE-19 cells were compared with the extracted human anterior capsule of the primary human LEC. No significant difference was observed across the three cell types (Figure 1C,D). Together with western blotting and qRT-PCR analysis, we found that CSPG5 protein (Figure 1E,F) and mRNA (Figure 1G) were significantly enhanced in the lens epithelium of SIC patients compared to that of patients with ARC (Figure 1F,G). These data suggested that CSPG5 was upregulated by dexa.

### 3.3. Effect of CSPG5 Downregulation on Dexa-Induced Cell Migration

To determine the appropriate dexa concentration for further experiments, we assessed cell viability upon administration of 0.1 mg/mL and 0.5 mg/mL dexa and found it to be unaffected for 48 h. However, cell morphology was significantly altered (Appendix A). To evaluate the function of CSPG5, the protein was knocked down by transfecting HLE-B3 cells with a *CSPG5* shRNA. The knockdown did not affect HLE-B3 cell viability for 48 h (Figure 2A). However, dexa-induced F-actin expression was suppressed in CSPG5 knocked-down HLE-B3 cells (Figure 2B). Furthermore, dexa-induced wound healing was significantly suppressed (Figure 2C). To understand this further, the epithelial–mesenchymal transition (EMT) markers related to cell migration were evaluated. The expression of fibronectin and CSPG5 was increased by dexa treatment, and this phenotype was suppressed by CSPG5 knockdown (Figure 2D–H). These results demonstrate that CSPG5 is involved in cell migration.

### 3.4. Dexa-Induced Expression and Translocation of Transcription Factors in HLE-B3 Cells

To identify the regulatory factors of CSPG5 in dexa-treated HLE-B3 cells, we focused on the transcriptional regulation of CSPG5. Previous reports on hepatocellular carcinoma indicated an association between the transcription factors EZH2 and B-Myb [17]. Therefore, the expression of these transcription factors was analyzed. Dexa did not affect the expression of EZH2 mRNA but increased B-Myb mRNA expression after 4 h of treatment after 24 h (Figure 3A–E). Comparative analysis of cytoplasmic and nuclear proteins revealed that dexa significantly increased the expression of EZH2 and B-Myb proteins in the nucleus (Figure 3F–H). Immunofluorescence assay further confirmed that the expression of both EZH2 and B-Myb was significantly increased by dexa in the nucleus (Figure 3I,J). These results suggest that the transcriptional regulators EZH2 and B-Myb are upregulated and translocated to the nucleus by dexa (). Additionally, dexa increased the protein expression of EZH2 and B-Myb.

### 3.5. Effect of CSPG5-Related Transcription Factor Downregulation on Dexa-Induced Cell Migration

To elucidate the relationship between these transcription factors and CSPG5 expression, HLE-B3 cells were treated with the inhibitors EI1 and A490 to inhibit EZH2 and B-Myb, respectively. Dexa-induced *CSPG5* mRNA expression was suppressed by the inhibition of EZH2 and B-Myb (Figure 4A). Dexa-induced protein expression of fibronectin and CSPG5 was suppressed by EZH2 inhibition (Figure 4B,C). Additionally, B-Myb inhibition had similar suppressive effects (Figure 4D,E). Furthermore, dexa-induced wound healing was significantly suppressed by the inhibition of EZH2 and B-Myb (Figure 4F–I). In summary, the inhibition of EZH2 and B-Myb suppresses dexa-induced cell migration mediated by CSPG5 via the regulation of *CSPG5* gene expression.

### 3.6. Dexa-Induced CSPG5 Expression in Primary Human LEC

To verify the dexa-induced CSPG5 expression in the human lens, the primary human LEC were cultured with dexa. Dexa significantly increased the protein expression of CSPG5 in cells cultured for 8 h (Figure 5A,B). Dexa also significantly increased the protein expression of CSPG5 with chondroitinase (Figure 5C,D). Furthermore, dexa increased the protein expression of both fibronectin and α-SMA, an EMT marker, in these cells (Figure 5E–G). These results indicate that dexa induces CSPG5 expression and affects cell migration in primary human LEC.

## 4. Discussion

SIC is regularly managed with cataract surgery, an invasive and expensive procedure [2]. In cells other than those constituting the lens, steroids have been shown to affect growth factor expression, thereby disrupting cellular homeostasis, although little information is available on their action on LEC [4,5,6,7]. Therefore, our results provide vital information on the molecular mechanisms regulating CSPG5 in LEC, which is related to SIC development, and offer further suggestions for the treatment of SIC.

In this study, we have identified the changes in the expression of growth factors in HLE-B3 cells following dexa treatment. Specifically, CSPG5 expression was higher in the dexa-treated group than in the control group. Glucocorticoids affect the systemic or local production of growth factors [5,13]. Thus, the destruction of growth factor homeostasis by glucocorticoids is inevitable in the development of SIC. Therefore, the epithelial growth factor (EGF)-like transmembrane domain of CSPG5 may play a role in SIC development. In addition, a study that verified an increase in *CSPG5* mRNA expression in Rpe65-/-mice reported that *CSPG5* mRNA is also expressed in the lens [12]. We found that CSPG5 was expressed in LEC of the anterior capsule collected from ARC and SIC patients; SIC led to a greater increase in *CSPG5* mRNA expression in LEC than ARC. Accordingly, CSPG5, a growth factor in the lens, was selected as a candidate for studying SIC development.

CSPG5 was downregulated in dexa-treated HLE-B3 cells, and migration of LEC was inhibited. Moreover, the expression of EMT marker proteins, which was enhanced by dexa treatment, was reversed upon knocking down *CSPG5*, indicating that CSPG5 is involved in the EMT of LEC. Similarly, a study that investigated CSPG5 and the prognosis of hepatocellular carcinoma indicated that CSPG5 is related to the transcription factors EZH2 and B-Myb [17], which reportedly play critical roles in EMT and cell cycle progression [18,19,20,21]. These transcription factors are known for their involvement in cell proliferation and differentiation, particularly in cancer cells. Reportedly, the mRNA levels of *EZH2* increase during the posterior capsular opacification in LEC [19,20], and EZH2 regulates EGF-mediated EMT in human LEC [20]. Dexa further increased the upregulation and translocation of EZH2 and B-Myb into the nucleus of LEC. When cells were treated with the inhibitors of EZH2 (El1) and B-Myb (A490), the increase in CSPG5 expression, which occurred as a consequence of dexa treatment, was suppressed. Therefore, EZH2 and B-Myb are predicted to act as transcriptional regulators in the pathological mechanism through which CSPG5 causes SIC.

Previous studies have reported that the proliferation of LEC declines by inhibiting EGF receptors, thereby effectively preventing posterior capsular opacification [22,23]. In addition, inhibiting the proliferation of LEC and their EMT is important to hinder SIC development [5,6], and EGF has been shown to promote EMT in lens epithelial cells [20]. Therefore, we hypothesized that CSPG5 exists as a transmembrane protein in LEC and is modified in its EGF-like domain by steroids. In this study, we successfully used primary human LEC of the anterior capsule collected from ARC and SIC patients during surgery and cultured them for further analysis. Following dexa treatment, CSPG5 expression increased compared to that in the control group, and the expression of EMT markers such as fibronectin and α-SMA simultaneously increased. Particularly, the knockdown of CSPG5 following dexa treatment resulted in reverting the increased expression of fibronectin. This suggests that steroids may increase the EMT of LEC in vivo, and CSPG5 may be involved in this process.

Finally, as seen in the clinical results of this study, SIC occurs at a younger age than ARC patients, and the prevalence of SIC is lower than ARC, which are well-known clinical characteristics of SIC [5], so the two groups were not compared equally in this study. However, posterior subcapsular cataract, a common type of cataract in SIC, is also a type often seen in ARC. Therefore, posterior subcapsular cataract in younger ages are more likely to be SIC, so the authors rather expect the age difference between the two groups to mitigate the confounding variable. In addition, dosage and duration of steroid usage was important in SIC development [5,24], but it was not controlled in the study between patients. Moreover, other systemic medication and disease which are associated with development of PSC might not be totally excluded in patients. Despite these limitations, it was confirmed that CSPG5 plays a role in SIC, and it is thought that prospective research will be needed in the future.

## 5. Conclusions

CSPG5 is a growth factor that is involved in increasing the EMT of LEC when exposed to dexa and enhanced in LEC of SIC patients. Our data indicate that EMT is induced in LEC by the nuclear translocation of EZH2 and B-Myb, which in turn result in the subsequent upregulation of CSPG5. Inhibition of both CSPG5 and its transcription factors suppresses EMT in LEC. However, further studies are required to elucidate the mechanisms by which CSPG5 regulates SIC and the other involved molecular members in the pathway. Taken together, our results suggest that CSPG5 could be a therapeutic target for the prevention and treatment of SIC, a possibility that is to be explored in the future. 

## Figures and Tables

**Figure 1 cells-12-01705-f001:**
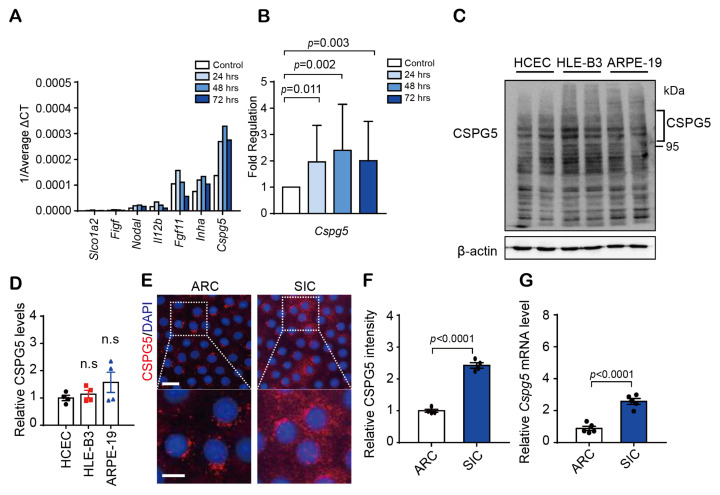
Effect on CSPG5 expression upon dexamethasone administration. HLE-B3 cells were cultured in a medium with dexamethasone (0.1 mg/mL) and analyzed at several time points (8, 24, 48, and 72 h). (**A**) Expression of growth factor genes was analyzed by RT2 Profiler PCR Array. (**B**) Expression of the *CSPG5* gene. Data are represented as the mean delta-CT values. Bars represent the mean ± % CV (*n* = 3). (**C**) Representative western blot image of CSPG5 expression in human corneal epithelial, HLE-B3, and ARPE-19 cells. (**D**) Graph showing the relative quantification of the protein expression levels of CSPG5 (*n* = 4). (**E**) CSPG5 protein expression in the human anterior capsule of the patients was detected by immunofluorescence (Red). DAPI was used as the nuclear marker (Blue). Scale bar, 50 μm. Scale bar of magnified image, 10 μm (Down). (**F**) Ratio of protein expression determined using ImageJ (*n* = 4). (**G**) Relative quantification graph of mRNA expression levels of *CSPG5* analyzed by qRT-PCR (*n* = 5). Bars represent the mean ± S.E.M. *p* < 0.05 was considered to be significant. ARC; Age-related cataract, SIC; Steroid-induced cataract.

**Figure 2 cells-12-01705-f002:**
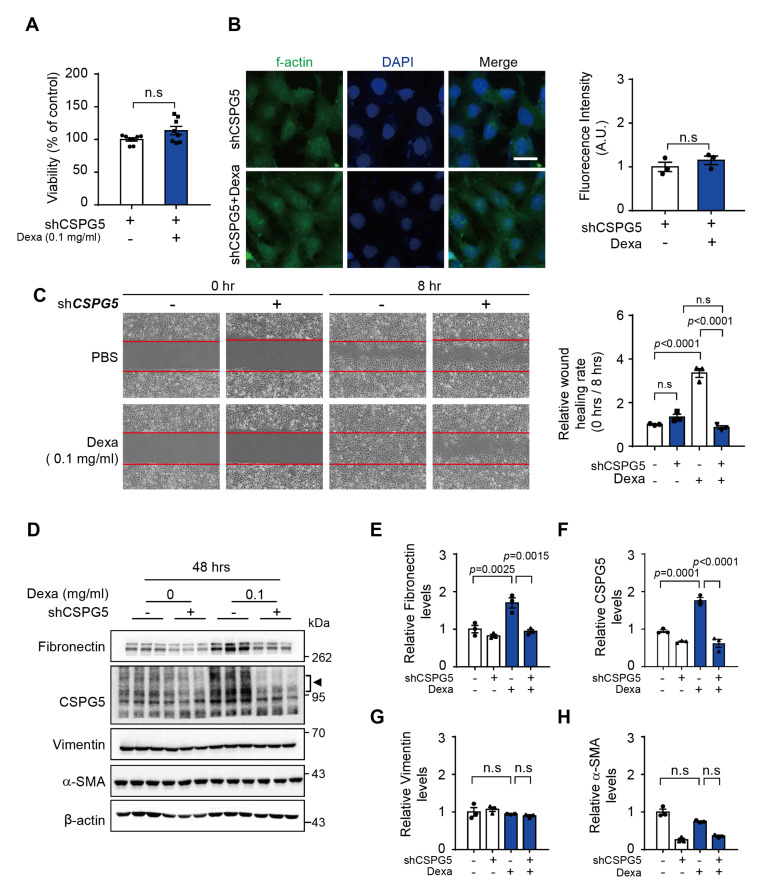
Effect of CSPG5 downregulation in dexa-induced cell migration. (**A**) CSPG5-knocked-down HLE-B3 cells were cultured in a medium with dexa (0.1 mg/mL) for 48 h. The effect of CSPG5 inhibition on the activity of mitochondrial dehydrogenases in dexa-treated HLE-B3 cells was measured by the CCK-8 assay (*n* = 8). The viability of control cells was set to 100%. Viability is expressed as a percentage of control cells. (**B**) Effect of CSPG5 inhibition on the cell morphology was detected by F-actin staining (Green). DAPI was used as the nuclear marker (Blue). Scale bar, 25 μm. Ratio of protein expression was determined using ImageJ (*n* = 3). (**C**) CSPG5-knocked-down HLE-B3 cells were cultured in a medium with dexa (0.1 mg/mL) for 8 h. The effect of CSPG5 inhibition on the dexa-induced cell migration was measured by wound healing assay. Relative healing rates are represented by the rate of the initial wound area at 0 h compared to 8 h (*n* = 3). (**D**) CSPG5-knocked-down HLE-B3 cells were cultured in a medium with dexa (0.1 mg/mL) for 48 h. Representative western blot image of dexa-induced EMT marker expression in *CSPG5*-knocked-down HLE-B3 cells. Arrowhead represents CSPG5. (**E**) Graph showing the relative quantification of protein expression levels of fibronectin, (**F**) CSPG5, (**G**) Vimentin, and (**H**) α-SMA (*n* = 3). Bars represent the mean ± S.E.M. *p* < 0.05 was the significance threshold.

**Figure 3 cells-12-01705-f003:**
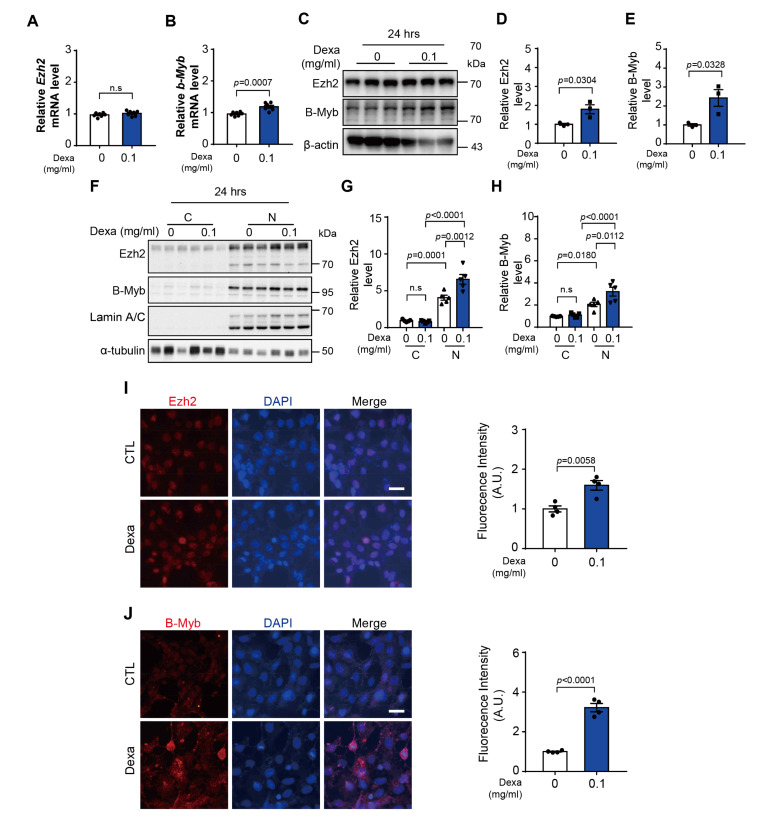
Effect on *CSPG5*-related transcription factors in dexa-treated HLE-B3 cells. (**A**) HLE-B3 cells were cultured in a medium with dexa (0.1 mg/mL) for 4 h. Graph showing the relative quantification of mRNA expression levels of *EZH2*, (**B**) *B-MYB* were analyzed by qRT-PCR (*n* = 7). (**C**) HLE-B3 cells were cultured in a medium with dexa (0.1 mg/mL) for 24 h. Representative western blot image of dexa-induced transcription factor expression in HLE-B3 cells. (**D**) Graph showing the relative quantification of total protein expression levels of EZH2 and (**E**) B-MYB (*n* = 3). (**F**) HLE-B3 cells were cultured in a medium with dexa (0.1 mg/mL) for 24 h. Representative western blot image of dexa-induced localization of transcription factors in HLE-B3 cells. (**G**) Graph showing the relative quantification of total protein expression levels of EZH2 and (**H**) B-MYB (*n* = 5). (**I**,**J**) HLE-B3 cells were cultured in a medium with dexa (0.1 mg/mL) for 48 h. The effect of dexa on transcription factor expression was detected by immunofluorescence (EZH2 and B-MYB; Red). DAPI was used as a nuclear marker (Blue). Scale bar, 25 μm. Ratio of protein expression determined using ImageJ (*n* = 3). Bars represent the mean ± S.E.M. *p* < 0.05 was the significance threshold.

**Figure 4 cells-12-01705-f004:**
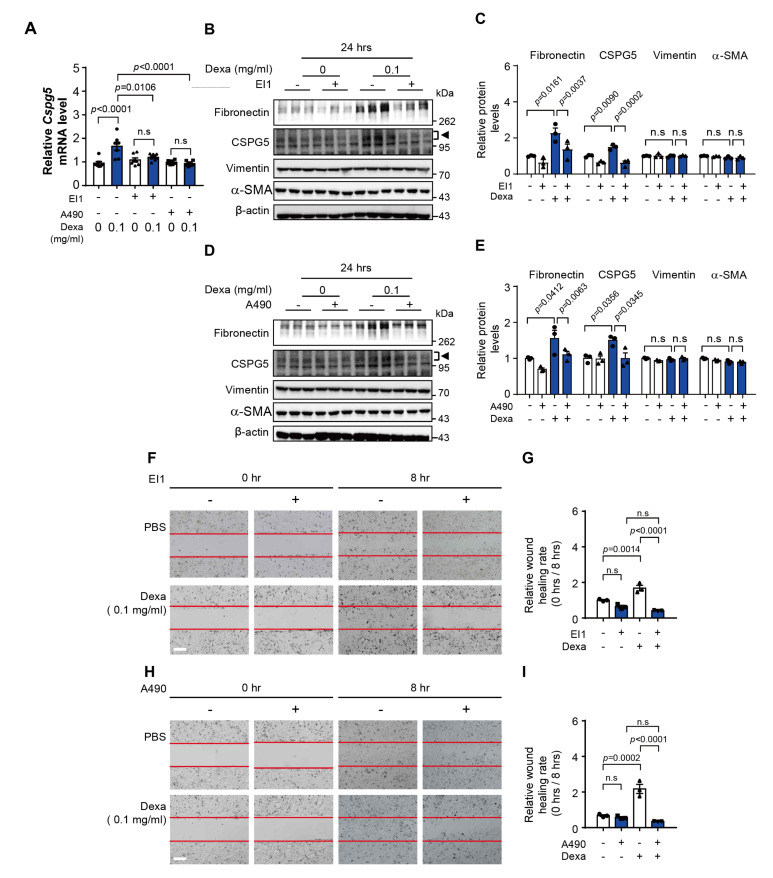
Effect of inhibition of *CSPG5*-related transcription factors in dexa-induced cells. (**A**) HLE-B3 cells were pretreated with EI1 and A490 (10 μM) and cultured in a medium with dexa (0.1 mg/mL) for 4 h. mRNA expression levels of *CSPG5* were analyzed by qRT-PCR (*n* = 7). (**B**) HLE-B3 cells were pretreated with EI1 (10 μM) and cultured in a medium with dexa (0.1 mg/mL) for 24 h. Representative western blot image of EMT marker expression in transcription factor-inhibited HLE-B3 cells. Arrowhead represents CSPG5. (**C**) Graph showing the relative quantification of protein expression levels of fibronectin, CSPG5, vimentin, and α-SMA (*n* = 3). (**D**) HLE-B3 cells were pretreated with A490 (10 μM) and then cultured in a medium with dexa (0.1 mg/mL) for 24 h. Representative western blot image of EMT marker expression in transcription factor-inhibited HLE-B3 cells. (**E**) Graph showing the relative quantification graph of protein expression levels of fibronectin, CSPG5, vimentin, and α-SMA (*n* = 3). (**F**) HLE-B3 cells were pretreated with EI1 (10 μM) and then cultured in a medium with dexa (0.1 mg/mL) for 8 h. The effect of EZH2 inhibition on the dexa-induced cell migration was measured by wound healing assay. (**G**) Relative healing rates are represented as a rate of the initial wound area at 0 h compared to 8 h (*n* = 3). (**H**) HLE-B3 cells were pretreated with A490 (10 μM) and then cultured in a medium with dexa (0.1 mg/mL) for 8 h. The effect of B-MYB inhibition on the dexa-induced cell migration was measured by wound healing assay. (**I**) Relative healing rates are represented as a rate of the initial wound area at 0 h compared to 8 h (*n* = 3). Data are represented as means ± S.E.M. *p* < 0.05 was the significance threshold. EI1, EZH2 inhibitor. A490, B-MYB inhibitor.

**Figure 5 cells-12-01705-f005:**
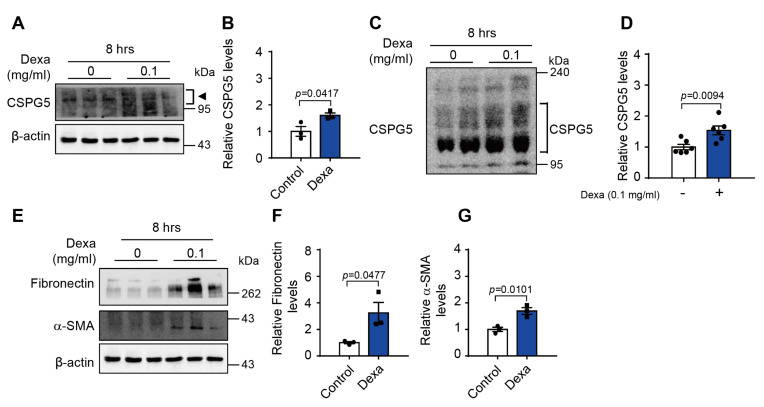
Effect on EMT markers in dexa-induced primary human lens epithelial cells. (**A**) Primary human lens epithelial cells were cultured in a medium with dexa (0.1 mg/mL) for 8 h. Representative western blot image of dexa-induced CSPG5 expression in primary human lens epithelial cells. Arrowhead represents CSPG5. (**B**) Graph of the relative quantification of CSPG5 protein expression (*n* = 3). (**C**) Representative western blot image of dexa-induced CSPG5 expression with chondroitinase. (**D**) Graph showing the relative quantification graph of protein expression levels of CSPG5 with chondroitinase (*n* = 6). (**E**) Representative western blot image of dexa-induced EMT marker expression in primary human lens epithelial cells. (**F**) Graph showing the relative quantification of protein expression levels of fibronectin and (**G**) α-SMA (*n* = 3). Bars represent the mean ± S.E.M. *p* < 0.05 was the significance threshold.

**Table 1 cells-12-01705-t001:** Demographic and clinical data of patients with cataract.

Variables		Age Related Cataract (*n* = 50)	Steroid Induced Cataract (*n* = 12)	*p*-Value
Sex (Male, %)		25 (50%)	8 (67%)	0.303
Age (yrs, mean ± SD)		68.1 ± 0.96	59.9 ± 0.37	0.008
BCVA (LogMAR, mean ± SD)		0.70 ± 0.446	0.69 ± 0.520	0.977
IOP (mmHg, mean ± SD)		14.1 ± 0.74	14.4 ± 0.84	0.732
LOCS III Grade (mean ± SD)	NO	3.7 ± 0.80	3.2 ± 0.58	0.023
	NC	3.7 ± 0.81	3.2 ± 0.58	0.029
	C	2.6 ± 0.07	1.6 ± 0.79	0.003
	P	0.5 ± 0.54	2.3 ± 0.65	<0.001

BCVA = best corrected visual acuity; IOP = intraocular pressure.

## Data Availability

All datasets generated and analyzed during this study are included in this published article and its Appendix A. Additional data are available from the corresponding author upon reasonable request.

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
