# Peer review of "Role of Chondroitin Sulfate Proteoglycan 5 in Steroid-Induced Cataract"

_cells, 2023, doi:10.3390/cells12131705_

Round 1
Reviewer 1 Report (Previous Reviewer 3)
I read the paper entitled ”Role of Chondroitin Sulfate Proteoglycan 5 in Steroid-Induced Cataract” very carefully again including the comments from the authors and concluded that the paper is now acceptable for publication in your journal. The authors considered all recommendations from reviewer.
Author Response
Response to Reviewer 1 Comments
Point 1 : I read the paper entitled ”Role of Chondroitin Sulfate Proteoglycan 5 in Steroid-Induced Cataract” very carefully again including the comments from the authors and concluded that the paper is now acceptable for publication in your journal. The authors considered all recommendations from reviewer.
Response 1 : Thank you for your positive comments. For more improvements, We added more information about SIC in introduction section, in which included characteristics of SIC development. We have revised the paper to add a more detail explanation in discussion section.
Reviewer 2 Report (New Reviewer)
Thank you for giving me an opportunity to review such a good article. This study has a meaning in firstly suggesting the potential of CSPG5 as a therapeutic target of steroid-induced cataract by well-designed in-vitro study. From cell culture and treatment to transfection, PCR, and Western blot analysis, through the thorough experimental design, readers would be able to understand the effects of CSPG5 on SIC in a cellular level.
Although the plan of the experiment and its results were not insufficient to support the conclusion, there are several other minor revision requests and curious point.
1. Firstly, a more detailed explanation of the steroid-induced cataract (SIC) would be helpful so that even people without ophthalmic knowledge can understand the SIC. For example, even steroid medicines are classified into five groups, which include glucocorticoids, mineralocorticoids, androgens, estrogens, and progestins, among them, only glucocorticoids are clinically known to produce steroid-induced cataract, characterized by posterior cataract. Also, this cataract formation is induced by glucocorticoids such as hydrocortisone and prednisolone in a dose dependent manner, but not by mineralocorticoids, androgen, estrogen and progestin. This information would also explain the reason why this study chose dexamethasone for treatment.
2. Secondly, in definition of cataract, SIC was defined as a cataract associated only with systemic or topical steroids that possess glucocorticoid activity and the migration of aberrant LEC from the lens equator to the posterior lens pole and limited central area of the posterior subcapsular region.
This approach is well established based on the definition and clinical practice, but it would be more helpful to explain why there were no criteria for duration or dosage of steroids. This is probably because the pathogenesis of SIC is also related not only to the duration and dosage of steroid but also to individual sensitivity. The difference in individual sensitivity to steroids can be a confounding variable in itself. Using steroid does not necessarily makes SIC, so I recommend the authors to mention this part as a limitation.
3. From a similar perspective, due to characteristics of patients taking steroids, other drugs that can cause various drug-induced cataracts are often prescribed together. It seems necessary to explain how the researchers excluded other drug-induced cataracts from SIC.
Author Response
Please see the attachment
Response to Reviewer 2 Comments
Thank you for giving me an opportunity to review such a good article. This study has a meaning in firstly suggesting the potential of CSPG5 as a therapeutic target of steroid-induced cataract by well-designed in-vitro study. From cell culture and treatment to transfection, PCR, and Western blot analysis, through the thorough experimental design, readers would be able to understand the effects of CSPG5 on SIC in a cellular level.
Although the plan of the experiment and its results were not insufficient to support the conclusion, there are several other minor revision requests and curious point.
Point 1 : Firstly, a more detailed explanation of the steroid-induced cataract (SIC) would be helpful so that even people without ophthalmic knowledge can understand the SIC. For example, even steroid medicines are classified into five groups, which include glucocorticoids, mineralocorticoids, androgens, estrogens, and progestins, among them, only glucocorticoids are clinically known to produce steroid-induced cataract, characterized by posterior cataract. Also, this cataract formation is induced by glucocorticoids such as hydrocortisone and prednisolone in a dose dependent manner, but not by mineralocorticoids, androgen, estrogen and progestin. This information would also explain the reason why this study chose dexamethasone for treatment.
Response 1 : Thank you for your helpful suggestions for improvement. We added more information about SIC in introduction section, in which included characteristics of SIC development. We have revised the paper to add a more detail explanation in introduction section.
Point 2 : Secondly, in definition of cataract, SIC was defined as a cataract associated only with systemic or topical steroids that possess glucocorticoid activity and the migration of aberrant LEC from the lens equator to the posterior lens pole and limited central area of the posterior subcapsular region.
This approach is well established based on the definition and clinical practice, but it would be more helpful to explain why there were no criteria for duration or dosage of steroids. This is probably because the pathogenesis of SIC is also related not only to the duration and dosage of steroid but also to individual sensitivity. The difference in individual sensitivity to steroids can be a confounding variable in itself. Using steroid does not necessarily makes SIC, so I recommend the authors to mention this part as a limitation.
Response 2 : Thank you for your kind comments. We totally agreed with your opinion. The definition of SIC was well-established in many studies and reports. In the definition, there are no mention about dose and duration of steroid usage. However, SIC was reported as increase with time and dose dependent manner. (Thorne JE, Woreta FA, Dunn JP, Jabs DA. Risk of Cataract Development among Children with Juvenile Idiopathic Arthritis-Related Uveitis Treated with Topical Corticosteroids. Ophthalmology. 2020 Apr;127(4S):S21-S26). Therefore, we agreed with your advice and add limitation in discussion section as your opinion.
Point 3 : From a similar perspective, due to characteristics of patients taking steroids, other drugs that can cause various drug-induced cataracts are often prescribed together. It seems necessary to explain how the researchers excluded other drug-induced cataracts from SIC.
Response 3 : Thank you for your recommendation. As you mentioned, other drug or systemic disease such as diabetes has a significant risk for PSC development. As seen in supplement Table 2, we excluded other systemic disease. However, this exclusion may have limitations, we added limitations of this study in discussion section and addressed future prospective study.

This manuscript is a resubmission of an earlier submission. The following is a list of the peer review reports and author responses from that submission.
Round 1
Reviewer 1 Report
This work has investigated the possible role of the transmembrane protein chondroitin sulfate proteoglycan-5 (CSPG5) in the formation of steroid-induced cataract (SIC) using cultured human lens epithelial cells (LECs) and lens epithelia from patients with SIC. Treatment of the human LECs with dexamethasone caused an upregulation of CSPG5, and CSPG5 expression was higher in lens epithelia from SIC patients compared to those from patients with age-related cataract. Although the results are not overwhelmingly clear and convincing, they do seem to indicate some role for CSPG5 expression in the formation of SIC. The work appears to have been conducted carefully, and the results should be of interest. The manuscript is generally well-written.
Reviewer 2 Report
Review for: “Role of Chondroitin Sulfate Proteoglycan 5 in Steroid-Induced Cataract”
In the present study, the authors investigate the role of CSPG5 in SIC. However, after carefully analyzing the manuscript, major flaws were identified that need to be thoroughly addressed prior to being considered for publication. I will focus on highlighting some of these issues, which I encourage the authors thoroughly address prior to submission.
- In table 1, aging was found to be significantly different between compared groups. As aging is associated with cataract development, these results are likely to impact the study's validity. Furthermore, there are errors in the percentages presented relative to the N.
- Western blots. Several issues were identified in western blots. The expression of housekeeping proteins was widely variable. No information was given regarding which antibody was used. The amount of protein loaded in the gels was not disclosed. CSPG5 blots were full of unspecific bands and the authors should tell us what is the expected size.
- Immunofluorescence images. If the tested antibody was the same one used in the western blots, how can the authors expect the signal to be specific to CSPG5?
- ShRNA assays. Controls need to be performed demonstrating that the levels of CSPG5 go down.
Reviewer 3 Report
I read the paper entitled ”Role of Chondroitin Sulfate Proteoglycan 5 in Steroid-Induced Cataract” very carefully and concluded that the paper is acceptable in the present form for publication in your journal. The topic of the article is interesting. CSPG5 may be response for steroid-induced cataract. The results of the study may contribute for new aproache for prevetion of steroid-induced cataract in future.
The nuber of patients with steroid –induced cataract is very small and represents only ¼ of age related cataract. So this must be included in the discussion part. It is also an important difference between both groups regarding the age. In the text that must be exposed. The authors shold proceede with further investigations and take comparision between groups which are equal in the number and age.
In table 1 the title should be changes. Demographic date can not include BCVA and IOP, I suggest: demographic and functional data among cataract patiernts.
Round 2
Reviewer 2 Report
Paper was previously rejected and problems were not adressed